# Long-Term Outcomes of Adult Patients with Homocystinuria before and after Newborn Screening

**DOI:** 10.3390/ijns6030060

**Published:** 2020-07-30

**Authors:** Kenji Yamada, Kazunori Yokoyama, Kikumaro Aoki, Takeshi Taketani, Seiji Yamaguchi

**Affiliations:** 1Department of Pediatrics, Shimane University Faculty of Medicine, 89-1, En-ya-cho, Izumo, Shimane 693-8501, Japan; ttaketani@med.shimane-u.ac.jp (T.T.); seijiyam@med.shimane-u.ac.jp (S.Y.); 2Secretariat of Special Formula, Aiiku Maternal and Child Health Center, Imperial Gift Foundation Boshi-Aiiku-Kai, 5-6-8, Minami Asabu, Minato-ku, Tokyo 106-8580, Japan; milk@boshiaiikukai.jp (K.Y.); aoki@boshiaiikukai.jp (K.A.)

**Keywords:** homocystinuria, cystathionine β-synthase deficiency, newborn screening, long-term outcome, social outcome, vitamin B_6_, methionine

## Abstract

Background: Homocystinuria (HCU) is a rare inherited metabolic disease. In Japan, newborn screening (NBS) for HCU (cystathionine β-synthase deficiency) was initiated in 1977. We compared the outcomes between patients detected by NBS (NBS group) and clinically detected patients (non-NBS group). Methods: We administered questionnaires about clinical symptoms and social conditions to 16 attending physicians of 19 adult HCU patients treated with methionine-free formula. Results: Eighteen patients (nine patients each in the NBS and non-NBS groups) participated. The frequency of patients with ocular, vascular, central nervous system, and skeletal symptoms in the NBS group was lower than that in the non-NBS group. Intellectual disability was observed in one and eight patients in the NBS and non-NBS groups, respectively. Concerning their social conditions, all patients in the NBS group were employed or still attending school, while only two patients in the non-NBS group were employed. Three of the four patients who discontinued treatment presented some symptoms, even in the NBS group. Conclusion: The social and intellectual outcomes of adult Japanese patients with HCU detected by NBS were favorable. However, even in the patients in the NBS group, some symptoms might not be preventable without continuous treatment.

## 1. Introduction

Homocystinuria (HCU) is a rare inherited metabolic disease characterized by the accumulation of homocysteine (Hcy) and its metabolites in the blood and urine [1,2]. HCU is classically categorized into three types depending on the specific enzymes involved in the metabolism of sulfur-containing amino acids that are deficient. The three types are as follows: (1) cystathionine β-synthase (CBS) deficiency (OMIM, 236200); (2) defect in cobalamin metabolism; (3) methylenetetrahydrofolate reductase deficiency. In CBS deficiency, the conversion of Hcy to cystathionine is impaired, and CBS deficiency is known as classic homocystinuria or homocystinuria type I [3]. The major clinical manifestations of CBS deficiency include the dislocation of the optic lenses, osteoporosis, “marfanoid” habitus, learning difficulties, and a predisposition to thromboembolism [4]. Because CBS deficiency is clinically heterogeneous and exhibits a wide range of outcomes, some patients have severe clinical phenotypes from childhood, while other patients may be asymptomatic until adulthood. Furthermore, CBS deficiency is classified into two phenotypes depending on vitamin B_6_ responsiveness. The B_6_-responsive type generally results in a milder phenotype [5].

Newborn screening (NBS) for HCU is performed in several countries and regions, including Western Europe, Australia, the United States of America, and Japan [6]. The prevalence varies widely by ethnicity and has been previously reported to range from 1:1800 to 1:1,000,000, with an overall estimated prevalence of 1:344,000 [7,8,9,10,11]. However, the true frequency is unknown and is thought to be higher than the prevalence detected by both NBS and clinical identification [9,12].

In Japan, NBS for HCU was initiated in 1977 and involves measuring methionine (Met) levels as the diagnostic marker. This strategy detects only CBS deficiency. The detection incidence of HCU (CBS deficiency) in Japan has been reported to be 1:800,000 to 1,000,000 births [13]. Early detection by NBS enables therapeutic intervention from early infancy and the maintenance of lower levels of blood Hcy, which may prevent the development of complications and consequently improve clinical outcomes in terms of both mortality and morbidity [14,15]. Although we reported the outcomes of inborn errors of metabolism in Japanese patients, including HCU detected by NBS, fifteen years ago [16], to date, the long-term outcomes of HCU are unknown. Here, we investigated the long-term outcomes in adult HCU patients, including clinically detected cases.

## 2. Materials and Methods

We sent questionnaires to 16 attending physicians of 19 adult patients with HCU who were older than 20 years as of October 2017 and who continued to be treated with a Met-free amino acid formula supplied by the Secretariat of Special Formula, Aiiku Maternal and Child Health Center. Patients with HCU types 2 and 3 were excluded from this study.

The questionnaires included questions regarding age, sex, clinical form, physical growth, prior NBS, onset age, first symptoms, metabolic data (i.e., blood methionine and Hcy levels), enzyme activity, genotype, treatments, symptoms (i.e., ocular involvement, vascular symptoms, central nervous system (CNS) disorders, and skeletal malformations), degree of intellectual disability, intermittent use of treatments, educational status, working status, marital status, and additional details. In this study, “marfanoid” included only skeletal symptoms, such as excessive height and/or arachnodactyly.

The subjects were divided into two groups, based on whether they underwent NBS as follows: (1) the NBS group was defined as those who underwent NBS after the initiation of NBS in Japan in 1977; (2) the non-NBS group included those who were born before 1977 or who were born in or after 1977 but did not undergo NBS. The results of the non-NBS group were compared with those of the NBS group; no statistical analysis was performed because of the small sample size in this study.

This study was approved by the Institutional Review Board of Shimane University in 28 September 2017 (#20170726-3).

## 3. Results

We received answers from 18 of 19 adult patients with HCU (response rate of approximately 95%). Ten males and eight females were included. Although 10 patients underwent NBS, one of these patients failed to be diagnosed with HCU based on NBS, and the diagnosis was made after the onset of symptoms. She was included in the non-NBS group. Eventually, the numbers of patients in the NBS and non-NBS groups were the same (9 per group).

### 3.1. Profiles of the Patients: Biochemical Findings and Treatments

The patient profiles are summarized in Table 1. The male/female ratio was 7/2 in the NBS group and 3/6 in the non-NBS group. The median ages were 25.8 years (21.3–36.7) and 44.3 years (32.2–59.2) in the NBS and non-NBS groups, respectively. Regarding vitamin B_6_ responsiveness, six and five patients in the NBS and non-NBS groups were non-B_6_ responsive, respectively. Elevated Met was found in a diagnostic test in 12 patients (NBS, eight patients; non-NBS, four patients) with available data. The median Met levels were 1264 μM (456 to 2433) and 565 μM (366 to 3903) in the NBS and non-NBS groups, respectively. Among the nine patients (NBS, five patients; non-NBS, four patients) with available data, the median blood Hcy levels were 71.7 μM (9.1 to 286) and 63.6 μM (22.2 to 292) in the NBS and non-NBS groups, respectively.

In three of the seven patients who underwent genetic testing, five types of variants—namely, p.H65R, p.G116R, p.G259S, p.M382R, and p.F531Gfs*9—in *CBS*, were identified. Only p.G116R was found in two patients. p.M382R was a novel mutation judged as “causing the disease” based on the Mutation Taster (http://www.mutationtaster.org/) results.

In the NBS group, all nine patients were treated with betaine and dietary therapy with Met-free formula, seven patients followed a protein-restricted diet, and six patients underwent antiplatelet therapy with aspirin. Additionally, three patients were treated with vitamin supplementation or anticoagulant therapy as other treatments. Meanwhile, in the non-NBS group, all nine patients received dietary therapy using a Met-free formula, and four patients followed a protein-restricted diet. Betaine and aspirin were administered to three patients. In the non-NBS group, one patient was treated for hypertension and hyperlipidemia, and another patient received antiplatelet therapy other than aspirin.

In total, four and two patients temporarily discontinued the treatment in the NBS and non-NBS groups, respectively. The reasons included economic problems, insufficient instructions by the physicians, and self-judgment mainly due to fewer subjective symptoms experienced during adulthood, difficulty in visiting distant hospitals, decreased motivation, the need for excessively strict control, and/or the cost of betaine. The discontinuation periods ranged from a few years to approximately ten years.

### 3.2. Clinical Symptoms

The major clinical symptoms in this survey are shown in Table 2. The NBS group had fewer complications than the non-NBS group. In particular, four of the nine patients in the NBS group were asymptomatic, and all patients were in their 20s.

Concerning optic involvement, retinal detachment was noted in one of the nine patients in the NBS group, who was aged 35 years, while among the nine patients in the non-NBS group, ectopia lentis, myopia, and glaucoma were observed in five, one and two patients, respectively.

Regarding vascular system complications, one patient in the NBS group had a thromboembolism in the pulmonary vessels at the age of 26 years, and another patient had cerebrovascular thromboembolic events at the age of 31 years. These two patients had a history of treatment interruption. In the non-NBS group, four patients developed thromboembolism in the period between the age of 6 years and their fourth decade of life, and one of them experienced complications of both cerebrovascular and pulmonary vascular obstructions.

Regarding CNS symptoms, psychiatric disability was noted in two of nine patients in the NBS group. One of these two patients with psychiatric disability also had intellectual disability. In contrast, all nine patients in the non-NBS group had some type of CNS symptoms, such as intellectual disability, epilepsy, psychiatric disability, and/or dystonia. Intellectual disability, including that of a mild degree, was observed in seven patients in the non-NBS group.

Regarding complications of the skeletal system, marfanoid was observed in three patients in the NBS group in their 30s. One patient also had osteoporosis. Meanwhile, all patients in the non-NBS group, except for one patient, presented some skeletal symptoms. Marfanoid, osteoporosis, scoliosis, and pectus excavatum were noted in five, two, five, and two patients, respectively.

Other complications, such as diabetes, arteriosclerosis obliterans, hyperlipidemia, and pneumonia, were noted in one patient each in the non-NBS group.

### 3.3. Life Outcomes: Intelligence, Education, Employment, and Other Outcomes

The outcomes in this survey are shown in Table 3. Physical development was within the normal range in six and three patients in the NBS and non-NBS groups, respectively. In total, two and four patients had excessive height in the NBS and non-NBS groups, respectively. Other physical findings, such as short stature and obesity, were noted in one and two patients in the NBS and non-NBS groups, respectively.

Regarding intelligence, eight patients showed normal intelligence, and one patient had mild intellectual disability among the nine patients in the NBS group. In contrast, only one of the nine patients in the non-NBS group demonstrated normal development, while three patients showed borderline, and one each had moderate and severe intellectual disability. An unknown degree of intellectual disability (may be moderate or severe) was noted in the other three patients in the non-NBS group.

Regarding education status, the final educational attainment was university, vocational school, high school, and school for the handicapped in two, four, one, and one patients, respectively, in the NBS group. In contrast, in the non-NBS group, the numbers of patients were two, zero, two, and three, respectively.

Regarding other outcomes, including employment and marital status, eight of nine patients in the NBS group were employed, and the other patient was still attending school. In the non-NBS group, two patients were employed, and three patients each were unemployed or living in a facility for the handicapped. Two patients in the NBS group were married, while no patient was married in the non-NBS group. Although all nine patients in the NBS group were alive, three patients in the non-NBS group had already died. The causes of death in these three patients were thalamic hemorrhage, pneumonia, and unknown. No further information regarding the three deceased patients was available.

In the other free description, all attending physicians stated the need for life-long treatment. Additionally, some physicians described the need to support medical expenses, including the treatment cost of betaine, and the necessity for consultation with internal medicine and psychiatry specialists.

## 4. Discussion

Our study revealed that NBS substantially contributed to the improvement in the long-term outcomes of Japanese patients with HCU but that the symptoms might progress even in patients detected by NBS. All patients in the NBS group worked or attended school, and all had normal mental development, except for one patient. NBS for HCU was previously reported to be an effective and recommended program [17]. The early start of treatment to maintain low levels of Hcy improved the long-term outcomes [18], and our study supports the finding that outcomes in Japanese patients are similar to those in previous reports.

Our results also indicate that lifelong continuous treatment is important to the achievement of improved long-term outcomes. Three of the four patients who discontinued their treatments presented some symptoms, even in the NBS group, while three of the five patients continuing treatments were asymptomatic. However, because information regarding the Hcy levels during the treatment period, including the discontinuation period, was not available in our study, the relationship between the outcomes and treatment interruption could not be fully elucidated.

Additionally, our results suggest that, even if HCU is detected by NBS and is continuously treated, the condition is likely to progress in some cases. In our study, the condition of the patients in their 30s seemed to be more severe than that of the patients in their 20s, even within the same NBS group. This finding might indicate that the management was improved or that their symptoms progressed. Our results also explored the responsiveness to treatments. Because marfanoid and psychiatric disability were observed in some patients in the NBS group, these symptoms might not be completely prevented even by early detection and intervention. However, scoliosis and pectus excavatum might be responsive to early intervention because these symptoms were present only in the non-NBS group. Ectopia lentis and intellectual disability also seemed to be responsive to early treatment.

Regarding vitamin B_6_ responsiveness, two of 11 patients could be considered B_6_-responders in our study, which is similar to a previous report revealing that 15% of Japanese HCU patients were vitamin B_6_ responders (based on a report in a domestic Japanese journal). The Japanese prevalence of B_6_ responders was higher than that in Ireland, where 1 in 25 patients was a B_6_ responders [18], while a report showed that 231 of 629 patients (36.7%) were vitamin B_6_ responders [19]. However, because not all Japanese patients, such as pyridoxine-responsive patients treated without Met-free formula, were included in our study, our results for vitamin B_6_ responsiveness might not correctly reflect the true prevalence in Japan.

Concerning the genetic background, it has been suggested that p.G116R might be common among Japanese patients, although genetic testing information was available for only three patients. In the European population, it has been reported that the p.I278T mutation is common and is associated with B_6_ responsiveness [9]. Additionally, the p.T191M, p.G307S, and p.R336C mutations are relatively common in some populations [8,20,21]. However, these mutations were not observed in the Japanese patients in our study.

Although 1 in 10 patients born between 1977 and 1997 was a false-negative case in our study, our results could not provide information about the sensitivity and specificity of screening tests using Met levels because of the small sample size. It is well known that the sensitivity and specificity of screening tests using Met levels alone are insufficient [22]. The selection of appropriate markers and setting of accurate cut-off values are future challenges for the NBS for HCU in the Japanese population.

There may be some limitations of our study. For example, patients treated with Met-free formula who could be traced by our institution were recruited for our study. Patients treated with only betaine and vitamin B_6_, patients who died before adulthood, and patients who interrupted treatment with Met-free formula during this survey, were not included. Information regarding the three dead patients was insufficient. Because approximately 30,000,000 babies were screened between 1977 and 1997 in Japan (1,200,000 to 1,800,000 births per year), there should, theoretically, be approximately 40 adult patients with HCU, but only 10 patients could be enrolled in our study. Therefore, our results do not correctly reflect the long-term outcomes of all Japanese patients with HCU. Furthermore, because HCU is a progressive disease, the differences in outcomes between the two groups may be associated with, not only NBS, but also age. We did not collect information to estimate the severity of disease except for symptoms; therefore, we could not compare disease severity between the two groups. Nevertheless, we believe that the severity is similar between the two groups because the Met and/or Hcy levels in the NBS group at diagnosis were similar to those in the non-NBS group. Therefore, we could not definitively conclude that the positive outcomes were all due to NBS. However, because there are few reports in which the long-term outcomes of patients with HCU detected by NBS were compared with those of clinically detected patients on the same scale and because the long-term outcomes of Japanese patients are unknown, our results are important for the investigation of the effect of NBS on HCU.

## 5. Conclusions

The long-term, particularly social and intellectual, outcomes of Japanese adult patients with HCU detected by NBS were favorable compared with those of patients with clinically detected HCU. However, even in the patients in the NBS group, some symptoms might not be preventable, and long-term outcomes may worsen if treatment is interrupted.

## Figures and Tables

**Table 1 IJNS-06-00060-t001:** Overview of the participating Japanese patients with HCU.

	NBS Group (*n* = 9)	Non-NBS Group (*n* = 9)
Sex		
Male	7	3
Female	2	6
Median age (range) (years)	25.8 (21.3–36.7)	44.3 (32.2–59.2)
Clinical form		
B_6_ responsive	2	0
Non-B_6_ responsive	6	5
Unknown	1	4
Diagnostic test		
Median Met (range) (μM)	1264 (*n* = 8) (456–2433)	565 (*n* = 4) (366–3903)
Median Hcy (range) (μM)	71.7 (*n* = 5) (9.1–286)	63.6 (*n* = 4) (22.2–292)
Number of patients undergoing tests for the urinary excretion of homocystine	7	5
Number of patients undergoing tests for CBS activity	2	2
Number of patients undergoing genetic tests	5	2
Treatments		
Met-free formula	9	9
Protein-restricted diet	7	4
Betaine	9	3
Aspirin	6	3
Dipyridamole	0	0
Others	3	2
Number of patients with a history of treatment interruption	4	2

Met, methionine; Hcy, homocysteine; CBS, cystathionine β-synthase.

**Table 2 IJNS-06-00060-t002:** Comparison of the major clinical symptoms in adult patients.

	Current Age	NBS Group (*n* = 9)		Non-NBS Group (*n* = 9)		
		20s (*n* = 5)	30s (*n* = 4)	30s (*n* = 4)	40s (*n* = 3)	50s (*n* = 2)
Eye	ectopia lentis	0	0	2	2	1
	myopia	0	0	0	1	0
	glaucoma	0	0	0	1	1
	other (retinal detachment)	0	1	0	0	0
Vascular system	coronal	0	0	0	0	0
	pulmonary	1	0	0	1	0
	cerebral	0	1	2	1	0
	other	0	0	0	0	1
Central nervous system	intellectual disability	0	1	2	3	2
	epilepsy	0	0	2	0	1
	psychiatric disability	0	2	1	0	0
	other (dystonia)	0	0	0	0	1
Skeletal system	marfanoid ^#^	0	3	2	3	0
	osteoporosis	0	1	1	1	0
	scoliosis	0	0	2	2	1
	pectus excavatum	0	0	1	0	1
	other	0	0	0	0	0
Other *		0	0	1	2	1
No symptoms		4	0	0	0	0

#: marfanoid involves excessive height and/or arachnodactyly. *: other symptoms included diabetes, arteriosclerosis obliterans, hyperlipidemia, and pneumonia in one patient each. The cumulative total number of patients was counted.

**Table 3 IJNS-06-00060-t003:** Comparison of clinical and social status.

	NBS Group (*n* = 9)	Non-NBS Group (*n* = 9)
Physical development		
Normal	6	3
Excessive height	2	4
Other *	1	2
Intelligence		
Normal	8	1
Borderline	0	3
Mild intellectual disability	1	0
Moderate intellectual disability	0	1
Severe intellectual disability	0	1
Unknown degree of intellectual disability (may be moderate or severe)	0	3
Education status		
University	2	2
Technical school	4	0
High school	1	2
Junior high school	0	0
School for handicapped	1	3
Unknown	1	2
Employment status		
Attending school	1	0
Employed	8	2
Unemployed	0	3
Living in a facility for the handicapped	0	3
Unknown	0	1
Marital status		
Married	2	0
Unmarried or not yet married	5	6
Outcome		
Alive	9	6
Dead	0	3

* Other includes obesity or a short stature.

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
