# Peer review of "Long-Term Outcomes of Adult Patients with Homocystinuria before and after Newborn Screening"

_2409-515X, 2020, doi:10.3390/ijns6030060_

Round 1

Reviewer 1 Report

This is a well-written article outlining longer-term outcomes for individuals with HCU detected on newborn screening as well as those detected outside of NBS. I do not have major concerns with the paper and recommend acceptance. The only confusing component for me was Table 1 - I didn't quite understand what 'intermittance of treatments' meant or what units the numbers were measuring. Additionaly, the 'number of tests' refers to how many times the patients were tested to confirm the diagnosis?

Author Response

Reviewer 1:This is a well-written article outlining longer-term outcomes for individuals with HCU detected on newborn screening as well as those detected outside of NBS. I do not have major concerns with the paper and recommend acceptance. The only confusing component for me was Table 1 - I didn't quite understand what 'intermittance of treatments' meant or what units the numbers were measuring. Additionaly, the 'number of tests' refers to how many times the patients were tested to confirm the diagnosis?

Thank you for your review and constructive comments. We are honored that you appreciate our manuscript. Our manuscript has been proofread again by native speakers, and all changes are shown in red. 

In our previous table 1, “Intermittence of treatments” meant the number of patients with HCU who had had a history of treatment interruption. Similarly, “the number of tests” meant the number of patients who underwent the diagnostic tests. Therefore, according to your comments, Table 1 was corrected.

Please also see the attachment.

Reviewer 2 Report

The incidence of Homocystinuria (HCU) is a very rare inborn error of amino acid metabolism and publication on the outcome of newborn screening (NBS) for this condition is of great value. Unfortunately the authors haven't provided sufficient details of the NBS cohort (n=9) given that the program was initiated in 1977 with over 1,000,000 screened. It is well know that the use of a methionine level to detect individuals with HCU isn't effective, when compared to PKU, lacking both sensitivity & specificity. Its is imperative that details of the NBS screened cohort are provided to fully appreciate the comparison of the clinical outcome of both groups. Whilst the author's cite an article by Aoki etal published in a journal not readily available, makes it even more important to provide the details of the NBS results. The clinical outcome was assessed in both NBS & non NBS screened groups at an age range of 21.3 - 36.2 & 32.2 - 59.2 years respectively the authors provide no additional comments on the relevance of this nor to the severity of the disease in each group.

Recommend that the authors consider providing these details to enhance the manuscript.

Author Response

Reviewer 2: The incidence of Homocystinuria (HCU) is a very rare inborn error of amino acid metabolism and publication on the outcome of newborn screening (NBS) for this condition is of great value.

  Thank you for your review and constructive comments. We are honored that you appreciate our manuscript. According to your feedback of our “English language and style”, our manuscript has been proofread again by native speakers, and all changes are shown in red.  

Unfortunately the authors haven't provided sufficient details of the NBS cohort (n=9) given that the program was initiated in 1977 with over 1,000,000 screened. It is well know that the use of a methionine level to detect individuals with HCU isn't effective, when compared to PKU, lacking both sensitivity & specificity.

  Our study has many limitations. One of them is the small number of participants. As mentioned in the discussion section of our previous manuscript, because approximately 30,000,000 babies were screened between 1977 and 1997 (1,200,000 to 1,800,000 births per year), and the prevalence was reported to be 1:800,000 births in Japan (as reported in Japanese domestic journal), there should theoretically be approximately 40 adult patients with HCU. However, only 10 patients were included in our study. Meanwhile, although one of 10 patients born between 1977-1997 was a false-negative case, our results could not provide information about the sensitivity and specificity of screening tests using Met levels. These points are emphasized in the revised manuscript.

Its is imperative that details of the NBS screened cohort are provided to fully appreciate the comparison of the clinical outcome of both groups. Whilst the author's cite an article by Aoki etal published in a journal not readily available, makes it even more important to provide the details of the NBS results. The clinical outcome was assessed in both NBS & non NBS screened groups at an age range of 21.3 - 36.2 & 32.2 - 59.2 years respectively the authors provide no additional comments on the relevance of this nor to the severity of the disease in each group.

  We agree that one of the limitations is a difference in the median age between the two groups. Therefore, we added “Furthermore, because HCU is a progressive disease, the differences in outcomes between the two groups may be associated with not only NBS but also age” to the last paragraph of the discussion section to clarify this point. Because we did not collect information to estimate the severity of disease except for symptoms, we could not compare disease severity between the two groups. However, we believe that the severity is similar between the two groups because the Met and/or Hcy levels in the NBS group at diagnosis are similar to those in the non-NBS group. We added this to the discussion as one of the limitations.

Recommend that the authors consider providing these details to enhance the manuscript.

  We apologize for not providing more details; however, the data provided in the previous manuscript were all the data we collected.

  Please also see the attachment.

Round 2

Reviewer 2 Report

The authors have addressed the comments made to the best of their knowledge and can confirm that the manuscript is now acceptable with the updated version.